# Different Trajectories for Diabetes Mellitus Onset and Recovery According to the Centralized Aerobic–Anaerobic Energy Balance Compensation Theory

**DOI:** 10.3390/biomedicines11082147

**Published:** 2023-07-30

**Authors:** Alexandre A. Vetcher, Kirill V. Zhukov, Bagrat A. Gasparyan, Pavel I. Borovikov, Arfenia S. Karamian, Dovlet T. Rejepov, Maria N. Kuznetsova, Alexander Y. Shishonin

**Affiliations:** 1Complementary and Integrative Health Clinic of Dr. Shishonin, 5 Yasnogorskaya Str., 117588 Moscow, Russia; kirizhuk@yandex.ru (K.V.Z.); b.gasparyan@shishonin.ru (B.A.G.); ashishonin@yahoo.com (A.Y.S.); 2Institute of Biochemical Technology and Nanotechnology, Peoples’ Friendship University of Russia, n.a. P. Lumumba (RUDN), 6 Miklukho-Maklaya St., 117198 Moscow, Russia; arfenya@mail.ru (A.S.K.); redzhepov-d@rudn.ru (D.T.R.); kuznetsova_mn@pfur.ru (M.N.K.); 3FSBI National Medical Research Center for Obstetrics, Gynecology and Perinatology n.a. V. I. Kulakov of the Ministry of Healthcare of the Russian Federation, 4, Oparina Str., 117997 Moscow, Russia; borpi@mail.ru

**Keywords:** HbA1c, arterial hypertension, blood pressure, diabetes mellitus, systolic peak, rhomboid fossa, central nervous system

## Abstract

We recently reported that the restoration of cervical vertebral arterial blood flow access (measured as systolic peak (PS)) to the rhomboid fossa leads to the recovery of the HbA1c level in the case of patients with a pre-Diabetes Mellitus (pre-DM) condition. The theory of centralized aerobic–anaerobic energy balance compensation (TCAAEBC) provides a successful theoretical explanation for this observation. It considers the human body as a dissipative structure. Reported connections between arterial hypertension (AHT) and the level of HbA1c are linked through OABFRH. According to the TCAAEBC, this delivers incorrect information about blood oxygen availability to the cerebellum. The restoration of PS normalizes AHT in 5–6 weeks and HbA1c in 12–13 weeks. In the current study, we demonstrate the model which fits the obtained experimental data. According to the model, pathways of onset and recovery from pre-DM are different. The consequence of these differences is discussed. The great significance of the TCAAEBC for medical practice forces the creation of an appropriate mathematical model, but the required adjustment of the model needs experimental data which can only be obtained from an animal model(s). The essential part of this study is devoted to the analysis of the advantages and disadvantages of widely available common mammalian models for TCAAEBC cases.

## 1. Introduction

Nowadays, the association of DM with AHT is ultimately considered a fact [1,2,3,4]. DM is a well-studied and exhaustively described disorder where the body either does not generate enough or respond normally to insulin, which causes blood glucose levels to be abnormally high, as well as its connections to cardiovascular diseases [5]. This is also correct for AHT—the very common health condition at which arterial blood pressure (BP) either in systolic, diastolic, or both points exceed the thresholds accepted by different medical communities [6,7,8]. Related to this, it needs to be underlined that there are different explanations for such connections, as well as proposals of different candidates on the roles of causes and consequences [9,10]. One of the most promising ways to explain this link is the opportunity provided by the TCAAEBC [11,12]. The main idea of it is the consideration of the patient as a system and the employment of a systematic approach to recovery. Some can remind that this idea was originally announced by Hippocrates in the form of “Treating the Patient, Not Just the Disease” [13]. Unfortunately, for the last four centuries, medicine has been developing rather under the influence of René Descartes’s theory of mind–body dualism [14,15]. Now, there is enough reason to create long-waiting reverence towards the systemic approach [16] which is offered by the TCAAEBC [17]. 

The TCAAEBC currently considers the human body from the point of the thermodynamics of irreversible processes, considering “good” and “bad” health conditions as positions on the map which describes the energetic behavior of the body. Since, according to the TCAAEBC, the body acquires NGSs as a result of adaptation to continuous misinformation due to OABFRH [18,19].

The typical example is the issue of the way a living organism balances between two methods of glycolysis—aerobic (AE) and anaerobic (AN)—to cover energetic requests. How does it work? A century ago, Ervin Bauer summarized that “All living organisms are characterized by being a system that is not in equilibrium in its environment and is so organized that it transforms the sources and forms of energy taken up from its environment into such state that acts against the establishment of equilibrium in the given environment. All the energy taken up by the organism from the environment must be fully used to deviate from the equilibrium state” [20,21]. This theory states that the organism within itself must possess available energy to prevail the equilibrium. It is required to maintain the necessary order. Otherwise, the organism is eventually destroyed by the external environment. Physics interprets this as an action of the first law of thermodynamics. It can be reflected, therefore, with the following equation:Q = U + W,
where Q is the heat, U is the internal energy of the system, and W is the work carried out by the system. 

The living body should keep a certain constant level of internal energy, ECONST. Then, for the living systems: E = ECONST + W,
where E is the sum of all energies flowing through the biosystem, ECONST is the abovementioned constant total internal energy of the biosystem, and W is the total work done by the biosystem in the external environment. It looks that to maintain a structure through which the constantly changing energy flow is passing on a constantly adjusted level of energy, we need feedback that regulates the consumption level. 

According to the TCAAEBC, to keep energy metabolism in the brain stem at a constant level of ECONST, the cellular and microcirculatory levels of the AN (glucose, lipoproteins, etc.—EAN) and AE (oxygen—EAE) molecules involved in it are under control to fulfill the equation:ECONST = EAE + EAN

The decrement in EAE causes two types of centralized adaptation reactions, which occur to maintain the level of ECONST. Such changes appear due to certain reasons, e.g., a decrement in oxygen content in the microcirculatory bed and brain stem cells. These are reactions of CAAEBC to maintain the level of ECONST. Since the reactions of AN compensation are less energy efficient, they are onset only after the complete exhaustion of the reserves of AE compensation reactions. The latter are neurogenic cardiovascular reactions, which are manifested as a steady increase in systolic BP (an increment in the cardiac output force), a constriction of the peripheral capillaries, and an increment in cardiac rate. The aim of the AE compensation reactions is an urgent increment in the brain stem blood perfusion and an eventual recovery of the EAE level. 

From the other side, AN compensation reactions appear to be neurohumoral which cause an increment in the AN metabolism of sugars, phospholipids, and other energetically rich biochemical compounds. The aim of these reactions is an increment in EAN to maintain, in the case of reduced EAE, the level of ECONST.

These reactions of the organism manifest phenotypic adaptation (PA) [22]. The PA of a living being to any changes in the environment starts with the aid of small forces through a less difficult itinerary. In the first step, oxygen saturation of the brain takes place, and then the reflex system which causes compensatory AHT is interrupted. If the brain encounters a lack of oxygen for an extended time, then according to the PA concept, changes take place at the biochemical level, i.e., the biochemical balance shifts, in other words, the AN part is increased and AE—decreased.

The brain, experiencing the lack of oxygen, considers its decrement as a decrement in the level of oxygen in the environment and tries to adapt the performance of the organism to the changed AE condition [23,24], i.e., the brain tries to accommodate the already changed, according to the information from the oxygen detector, conditions. But they have remained the same. Then, the brain starts to receive signals about the heart’s critical condition, and it should start to behave according to its designation as a control center. To save the cardiac resource, it adjusts the biochemical system to accommodate the reduced partial pressure of oxygen. Such an offset in the AE–AN equilibrium to AN maintains the overall balance of energy that is necessary to fulfill Bauer’s universal principle of biology and to balance the effect on the body of the second law of thermodynamics [20,21].

The best clinical example of the “fast” and “slow” compensatory mechanisms is squeezing the vessels of the neck to provoke a person’s short-term hypoxia (oxygen deprivation). The same can be obtained with tough physical exercise (sudden rise in oxygen consumption). Both cases instantly result in a reflexive increment in BP and heart rate [23,24]. After deprivation is over, all vital signs quickly return to normal. This is the case of “quick” adaptation.

The case of “slow” adaptation appears in the case of a long-term obstruction of the vessels due to, e.g., a narrowing of the lumen of the vessels due to the atherosclerotic process or cervical osteochondrosis, etc. Here, we observe the shift in the AE–AN balance. The best example is the development of the metabolic syndrome, in particular, DM.

It needs to be underlined that the TCAAEBC introduces to the description of AHT onset such terms as ECONST, which allows us to structure the balance of body energies, and EAE and EAN to describe AE and AN contribution to the overall energy flow of the body. In the TCAAEBC, we connect the partial pressure of oxygen in the brain stem as an index causing the brain to regulate these energy processes.

The idea that some cases of AHT are results of OABFRH by intervertebral disc compression with hernias and protrusions of the cervical spine leads to certain consequences. The anatomical features of the cervical vertebrae are such that veins and arteries pass through the holes in their transverse processes (*arteria vertebralis*, *venae vertebralis*). Due to the offset of the vertebrae (or disc), a deep muscle spasm around them appears. It then causes OABFRH. It is the PS reduction of up to five times which results in the dramatic reduction in the amount of delivered oxygen to the oxygen detectors in the brain stem. Since the detectors’ signal causes the brain to acknowledge the lack of oxygen, it takes emergency measures and forces the heart to increase its strength and/or heart rate so that blood, through all the blocks and obstacles, is still able to reach the brain and provide much-needed oxygen. The increment in pressure and/or heart rate which is needed to protect the brain from hypoxia is being developed. *Vice versa*, after unlocking the vertebral arteries, the pressure and heart rate should return to normal. Therefore, since patients with AHT have been healed through the restoration of vertebral arteries PS, then the TCAAEBC should be considered as confirmed. This is easy to register using measurements of BP and PS. The elimination of OABFRH during the correction of deep neck muscles leads to the measurable stable restoration of PS of *sinistra* and *dextra arteria vertebralis* to the normal values. To prove this, we need to compare BP and PS data on the animal model both before the introduction of OABFRH and after removal.

So far, this approach has been successfully employed in the explanation of AHT [14], DM [11], left ventricular hypertrophy [25], sudden cardiac death [26], vertebral cartilage issues [27], and problems with posture [28] (Figure 1).

Let us underline that the TCAAEBC provides the view on the way of interconnection of multiple NCOs through the acknowledgement of the main role of the central nervous system (CNS). From this perspective, the recent reports in connection(s) through the CNS of, e.g., autoimmune (one of the examples of which is DM) and neurodegenerative diseases [29] can be united on the level of physiology until molecular level is under clarification.

The visible success of the logical explanation leads to the next step—mathematical modeling, which allows testing and demonstrating critical points of verification [30]. To compile a model, it is necessary to start from the list of parameters to evaluate [31,32]. The most logical parameter, approved by WHO since 2006, to describe the degree of DM is hemoglobin-glycated hemoglobin (also known as glycosylated hemoglobin, HbA1c, and hemoglobin A1c (A1C)) [33,34]. The American Diabetes Association (ADA) 2015 chose the HbA1c level (H) ≥ 6.5% (48 mol/mol) as the diagnostic criterion for DM. For pre-DM, it should be from 5.7 to 6.4% [35].

H allows the performance of long-term glycemic control since it reflects the cumulative glycemic history of the preceding 3 months or so. It characterizes the average glucose level and demonstrates high reproducibility even with different methods of analysis [36]. H allows the measurement of chronic hyperglycemia and estimates the risk of long-term diabetes complications. In addition, H is the most abundant (>90%) glycated hemoglobin.

H is calculated according to Equation (1):(1)$$H=\frac{\mathrm{HbA}1c}{\mathrm{HbA}1c+\mathrm{HbA}0}\times 100\%$$
where HbA0 is the rest of the hemoglobin.

The current study is devoted to the modeling of H trajectories during onset and recovery from pre-DM in the case when it is caused by the events described by the TCAAEBC.

## 2. Materials and Methods

For our modeling, we used the previously reported data shown in Table 1, which demonstrates the absence of significant gender differences. Therefore, we used parameters from the combined sample.

## 3. Modeling

The mathematical model of the recovery from the adaptive OABFRH state of the body during the restoration of PS provides additional issues yet to be considered. As mentioned above, long-term adaptation occurs with repeated or ongoing OABFRH. It is a combination of structural, functional, and metabolic adaptive reactions. Therefore, the main feature of the model is taking into account a large number of such reactions that have different effects and effects over time. In model building, we used an approach similar to [39].

### Model Explanation

The adaptation processes are formed continuously as a result of repeated activation of the mechanisms of urgent adaptation to OABFRH. Homeostasis on each step generates conditions for optimal vitality of the organism. Eventually, a dissipative structure is formed with at least two stable points (attractors of the unstable focus type). The first point is the basic state of the body in the absence of OABFRH (Figure 2). The second point demonstrates the state of the body in the case of adaptation to OABFRH. The most important observation is that the onset and recovery itineraries are different.

In the real space of values, we define the functions of apparent OABFRH H (t, PS) and BP (t, PS), where t is time. Consider the dynamical system of evolution ∂/∂t of the functions H and BP; we denote r = PS for brevity. To account for the contribution to the general system of the cumulative effect of the action of each term H and BP, we interpret the coefficients DH and DBP as the values of their “range” with diffusion term $\frac{\partial^{2}}{\partial r^{2}}$ functions, according to the Equations (2) and (3):(2)$$\frac{\partial H}{\partial t}=a_{11}H+a_{12}BP+F\left(H,BP\right)+D_{H}\frac{\partial^{2}H}{\partial r^{2}},$$
(3)$$\frac{\partial BP}{\partial t}=a_{21}H+a_{22}BP+G\left(H,BP\right)+D_{BP}\frac{\partial^{2}BP}{\partial r^{2}}$$

The terms *F* and *G* have the meanings of the regression dependencies that may appear during field experiments (below, *F* = *G* = 0). The initial boundary conditions are *BP* (t,20) = 130 and *H* (0,*r*) = 100(1 − *r*/20).

The values are taken as typical parameters of the patient’s ultrasonography. The maximum PS does not exceed 45 cm/s and the time is 12 weeks. Later, the system is reduced to a dimensionless form and numerically solved using the FEM method implemented in the Wolfram Mathematica 13.0 environment. In Figure 3, one can see the relationships between H and PS. The most important observation from it is that the itinerary consists of two parts with a sharp change in the slope. This can be checked by more regular PS measurements in further studies on animal models that allow data collection with the desired periodicity. We especially left the axes without any scale since it should be defined after experimental data collection.

The modeling of DM-related and AHT-related parameter interactions opens a new page in TCAAEBC studies. There is a wide variety of blood flow parameters which are associated with arterial blood access to the rhomboid fossa that are relatively easy to determine during triplex sonography [40,41,42] and their association with H(t) is easy to evaluate. Our experience suggests that it is still possible to find the parameter that could be optimal for the characterization of OABFRH according to the TCAAEBC. Finding such a parameter could allow the control of recovery to be easier and faster. We are sure that the search should be continued. Moreover, nowadays there are additional methods proposed for quantitative blood flow measurements such as pulsed wave Doppler studies, high-frame-rate ultrasound vector flow imaging, phase contrast MRI, and several others [42,43,44]. The long list of proposed approaches looks endless, even if the number of proposed measurement parameters is much shorter.

The same is true with the understanding of the parameters that could be offered instead of HbA1c for better characterization of DM [45]. Since its acceptance by WHO as a major index for DM, its usefulness is regularly questioned [46,47]. A similar situation is with the parameters to characterize AHT [48,49,50].

## 4. Appropriate Animal Model(s)

The overview on the planned experimental setup is demonstrated in Figure 4.

Let us list the available and acceptable options from the medical community’s animal models with the features that are essential for the TCAAEBC verification (Table 2). Acceptance of the model is extremely important, since otherwise we could step into the same trap as Mendel, who was proposed to repeat his experiments on Hieracium because his model—Pisum sativum—looked less acceptable to his pen pal [51,52]. Let us underline that neither of these were the animal model.

From the very beginning, we would like to exclude primate models from consideration due to:

The absence of defined knowledge that required data could be obtained only from them;

The existing restrictions on the involvement of such models in nonobligatory experiments [53,54].

**Table 2 biomedicines-11-02147-t002:** Comparison of mammalian models on required-to-check CAAEBC parameters [55,56,57,58,59,60].

Model Animal	Results Are Transferrable to Human Clinical Situation(s)	Lack of Reserve Arterial Way to Rhomboid Fossa	General Convenience to Measure BP	Blood Boost for Biochemical Analysis	Simplicity of Brachycephalic Arterial Blood Flow Measurement	Availability of Reported Positive Experience with the Reverse Brachycephalic Arteries Blocking
Mice	+	−	+	+	+	−
Rats	+	−	+	+	+	+
Rabbits	+	−	+	+	+	−
Minipigs	+	+	+	+	+	−
Goats	−	−	+	+	+	−
Sheep	−	−	+	+	-	−
Guinea pigs	−	−	+	+	+	−
Cats	+	+	+	+	+	−
Dogs	+	+	+	+	+	−

There are some general points about aspects that should be taken into consideration to choose an appropriate animal model. Some experimentalists even underline that there is no ideal model for every clinical situation, but they can be more or less accurate to represent them [61]. Importantly, the contemporary knowledge of the detailed characteristics of each animal model is unavailable. Therefore, experiments often need to be designed on pure guess. This can be a problem later in the translation stage of the acquired data to human clinical situations. In the majority of the convenient animal models, the small diameter and tangled geometry often close the door to certain models [62].

The fundamental advantage of the animal model is the ease with which the results may be applied to clinical settings including humans [63,64]. It is frequently so difficult to identify the best model for a given situation that a pilot experiment is required [65,66].

The arterial blood supply in rats exhibits heterosegmental organization [59]. This fact is not in favor and has the greatest difference from real-world clinical scenarios of any model on the list. However, this is so far the only model where the elevation in BP was demonstrated after implementation of the plastic implants, causing atlantoaxial misalignment. The removal of the implant reversed BP to normal values. The evaluation of PS was not performed [24].

Let us think about the (mini)pig model’s drawbacks. Although there is no evidence on cervical vertebral arteries, there are reports that this model can create collateral blood flow to duplicate vertebral arteries in the situation of occlusion [67]. The fundamental benefit of using a minipig model to test CAAEBC is that the physiology and vascular anatomy are similar to those of people [68]. This model’s popularity is primarily due to similar clinical conditions.

When it comes to dogs and cats, these models show some similarities to the human artery system, which is helpful for our purposes. For example, rabbits are used for various cases of cardiovascular modeling [69,70,71]. However, the public’s attitude is significantly less positive about these models in comparison with the two listed above [72]. Therefore, we consider them as the last resort.

The abdominal aorta of the rabbit, used as a model, provides homosegmental blood flow. This model has a significant benefit in that it nearly completely lacks an intraspinal collateral arterial system [71]. The key benefits are cheaper maintenance expenses and simpler manipulation.

More segmental blood is present in the guinea pig model. The small segmental arteries are to blame for this [71]. This model is the least advantageous to model CAAEBC because of the inability of blocking blood passage to the rhomboid fossa due to the larger segmental arterial supply.

The mouse model is comparable to those using other rodents [73]. Despite all of this model’s benefits, including how quickly experimental symptoms can appear, how inexpensive it is, and how simple it is to manipulate [61], we believe it is flawed for the same reason as the rat model. Detailed information on the anatomy of the cervical part of the spinal cord vascular organization is extremely important to plan the experiment properly [74].

It looks like even if the rat model is already confirmed as a primary candidate because of its reported performance, the minipig and rabbit models could be considered as the second and third options.

## 5. Gaps in Evidence and Future Directions Instead of Conclusions

Even if the primary parameter for DM characterization is HbA1c (according to WHO), which does not allow for characterization faster than 90 days (according to its nature), to allow daily modeling, we need to collect data on daily glucose profile.

The observation of the behavior of the model describing the recovery from DM as a function of the restoration of the arterial blood flow access to the rhomboid fossa leads to the following outcomes:The onset and recovery itineraries from pre-DM are different;The slope of the HbA1c dependence on PS demonstrates two different areas. Each of them should have different health condition situations, which should be verified on animal models. These preliminary conclusions should be preceded by the optimization of the list of parameters to obtain a further description of OABFRH’s influence on DM. The collected data could be used in the evaluation of the set of proper parameters for the aging index(es). This became critically important in 2022, when WHO (in ICD-11) eventually started to consider aging as a health condition and not as a normal stage of development [75]; therefore, AHT and DM, which were considered aging satellites, could experience a change in mankind’s attitude toward them.

## Figures and Tables

**Figure 1 biomedicines-11-02147-f001:**
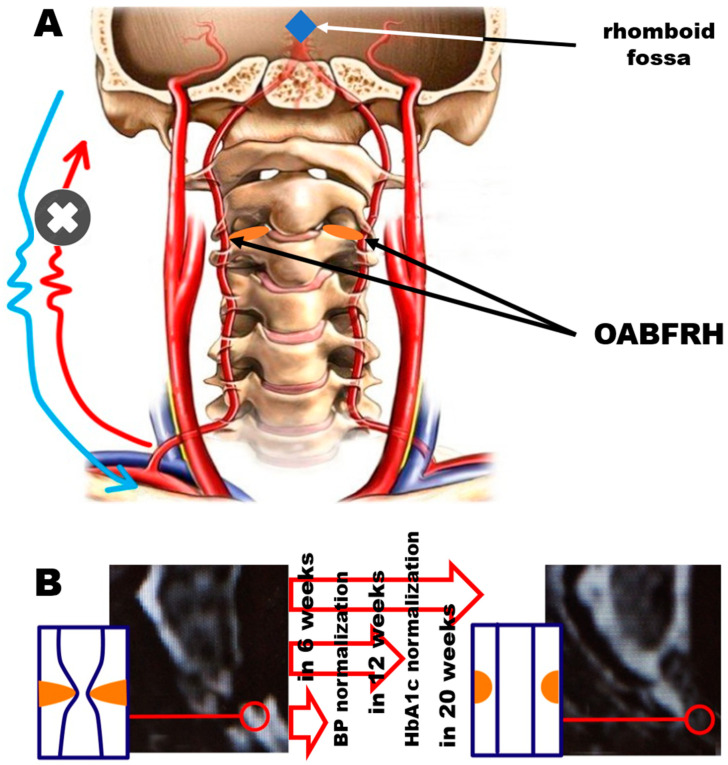
(**A**) Schematic of the TCAAEBC view on OABFRH. (**B**) Itinerary of patients’ recovery during the treatment according to [27].

**Figure 2 biomedicines-11-02147-f002:**
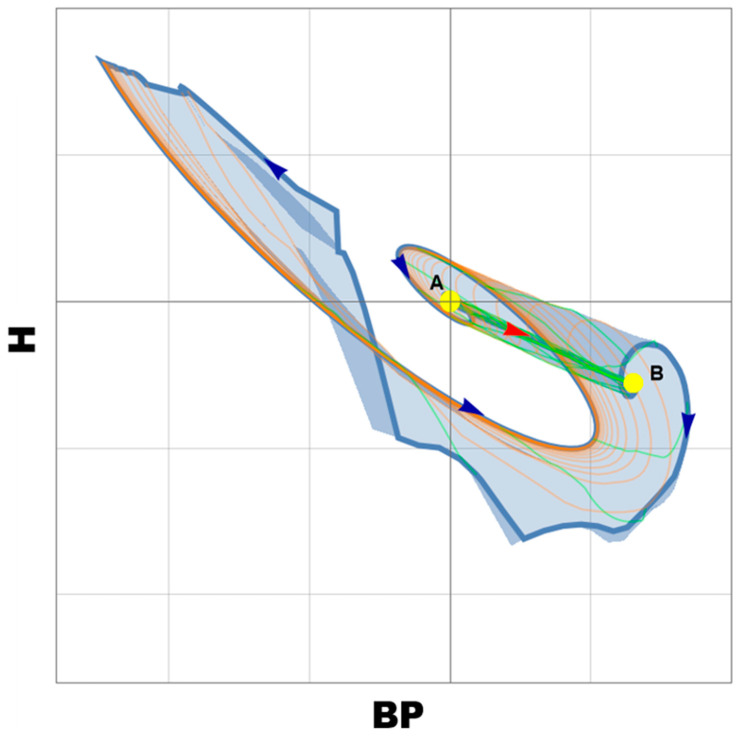
Point B is an unstable focus that appears during OABFRH and point A is an unstable focus in the absence of OABFRH of the dynamic system. The pathways from A to B are marked with dark arrows which show the direction.

**Figure 3 biomedicines-11-02147-f003:**
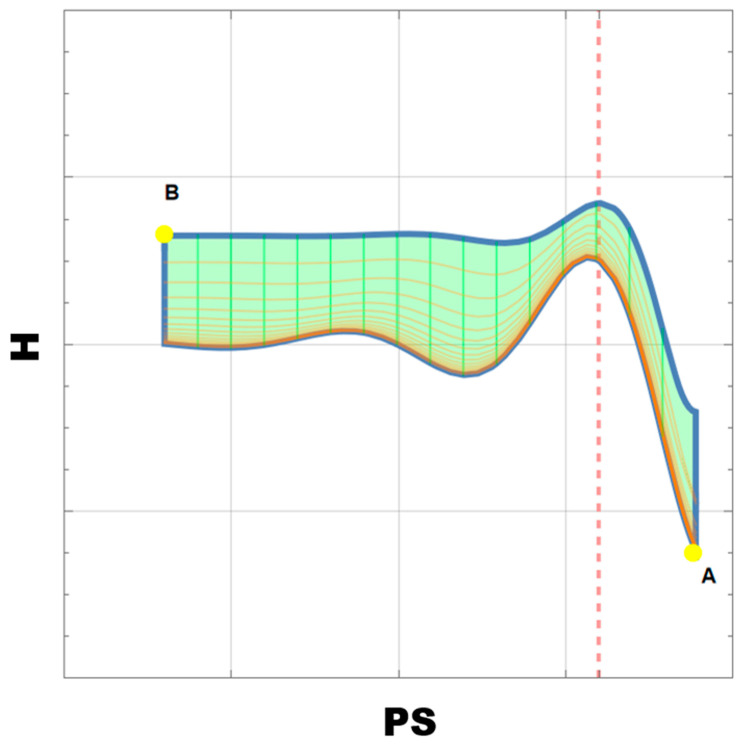
Function H (t,PS). A and B as described above.

**Figure 4 biomedicines-11-02147-f004:**
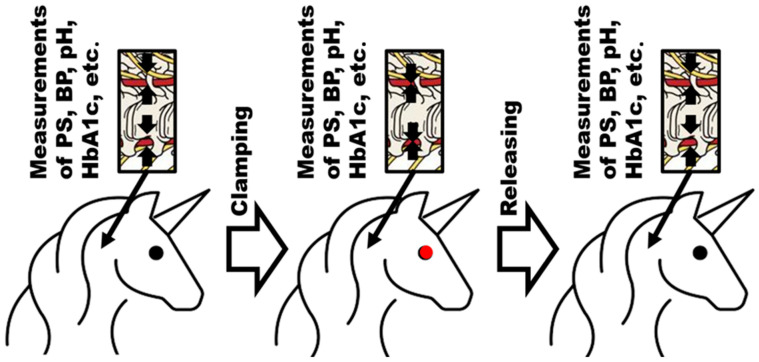
The animal model to check the CAAEBC applicability to AHT.

**Table 1 biomedicines-11-02147-t001:** Changes in BP, PS, and H from the above-described treatment (edited from [11]).

Parameter	M	F	M + F	Normal
Sample size	19	29	48	
Age, years	63.1 ± 11.7	65.5 ± 12.2	64.3 ± 12.0	
BP before treatment, torr	159.5 ± 18.3	163.5 ± 17.9	161.9 ± 18.1	
BP after treatment, torr	132.3 ± 19.2	131.7 ± 16.6	132.9 ± 17.3	<140 [37]
PS before treatment, cm/s	22.5 *±* 8.1	21.9 *±* 9.3	22.2 *±* 7.5	
PS after treatment, cm/s	41.7 *±* 6.7	43.2 *±* 7.4	42.5 *±* 7.8	48 *±* 10 [38]
H before treatment, %	6.03 ± 0.34	6.11 ± 0.45	6.08 *±* 0.41	
H after treatment, %	5.7 ± 0.63	5.73 ± 0.51	5.72 *±* 0.58	5.7–6.4 [35]

## Data Availability

Not applicable.

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
