# Peer review of "Different Trajectories for Diabetes Mellitus Onset and Recovery According to the Centralized Aerobic–Anaerobic Energy Balance Compensation Theory"

_biomedicines, 2023, doi:10.3390/biomedicines11082147_

Round 1
Reviewer 1 Report
Vetcher et al. present an article regarding the different trajectories for the Diabetes Mellitus onset and recovery according to the centralized aerobic-anaerobic energy balance compensation theory. Although the topic is interesting and the manuscript well-written, some considerations need to be clarified.
1. Introduction: General description of pre-diabetes mellitus, diabetes mellitus and the role of HbA1c is lacking. Please, add this general description and the following reference: “Cosentino F, et al.; ESC Scientific Document Group. 2019 ESC Guidelines on diabetes, pre-diabetes, and cardiovascular diseases developed in collaboration with the EASD. Eur Heart J. 2020 Jan 7;41(2):255-323. doi: 10.1093/eurheartj/ehz486"
2. Introduction: General description of arterial hypertension considering the latest ESC/ESH Guidelines for the management of arterial hypertension is lacking. Please, add this general description and the 3 following references:
1. “Williams B et al. ESC Scientific Document Group. 2018 ESC/ESH Guidelines for the management of arterial hypertension. Eur Heart J. 2018 Sep 1;39(33):3021-3104. doi: 10.1093/eurheartj/ehy339”.
2. "Perrone V et al. Treatment with Free Triple Combination Therapy of Atorvastatin, Perindopril, Amlodipine in Hypertensive Patients: A Real-World Population Study in Italy. High Blood Press Cardiovasc Prev. 2019 Oct;26(5):399-404. doi: 10.1007/s40292-019-00336-2".
3. Mancia Chairperson G et al. 2023 ESH Guidelines for the management of arterial hypertension The Task Force for the management of arterial hypertension of the European Society of Hypertension Endorsed by the European Renal Association (ERA) and the International Society of Hypertension (ISH). J Hypertens. 2023 Jun 21. doi: 10.1097/HJH.0000000000003480
3. A specific paragraph on “Gaps in evidence and future directions” should be added, especially in view of an application in clinical practice.
Author Response
2023-07-25
To Reviewer 1
Dear Reviewer:
Thank you so much for you high evaluation of our humble contribution as well as for your efforts to improve its readability. As about your comments, let me address them in the order in your review:
- Introduction: General description of pre-diabetes mellitus, diabetes mellitus and the role of HbA1c is lacking. Please, add this general description and the following reference: “Cosentino F, et al.; ESC Scientific Document Group. 2019 ESC Guidelines on diabetes, pre-diabetes, and cardiovascular diseases developed in collaboration with the EASD. Eur Heart J. 2020 Jan 7;41(2):255-323. doi: 10.1093/eurheartj/ehz486"
Done
- Introduction: General description of arterial hypertension considering the latest ESC/ESH Guidelines for the management of arterial hypertension is lacking. Please, add this general description and the 3 following references:
- “Williams B et al. ESC Scientific Document Group. 2018 ESC/ESH Guidelines for the management of arterial hypertension. Eur Heart J. 2018 Sep 1;39(33):3021-3104. doi: 10.1093/eurheartj/ehy339”.
- "Perrone V et al. Treatment with Free Triple Combination Therapy of Atorvastatin, Perindopril, Amlodipine in Hypertensive Patients: A Real-World Population Study in Italy. High Blood Press Cardiovasc Prev. 2019 Oct;26(5):399-404. doi: 10.1007/s40292-019-00336-2".
- Mancia Chairperson G et al. 2023 ESH Guidelines for the management of arterial hypertension The Task Force for the management of arterial hypertension of the European Society of Hypertension Endorsed by the European Renal Association (ERA) and the International Society of Hypertension (ISH). J Hypertens. 2023 Jun 21. doi: 10.1097/HJH.0000000000003480
Done
- A specific paragraph on “Gaps in evidence and future directions” should be added, especially in view of an application in clinical practice.
Done. We unite it with the Conclusions Section.
Please, let us know if we can do something else to improve our submission.
Regards
Dr. Alexandre A. Vetcher
Reviewer 2 Report
Alexandre A. Vetcher and colleagues present a quality and well-written experimental article focused on different trajectories for the Diabetes Mellitus onset and recovery according to the centralized aerobic-anaerobic energy balance compensation theory.
Authors suggest that the theory of centralized aerobicanaerobic energy balance compensation provides successful theoretical explanation for this observation. They considers the human body as a dissipative structure. They reported connections between arterial hypertension and the level of HbA1c are linked through the OABFRH. According to TCAAEBC, this delivers incorrect information about blood oxygen availability to the cerebellum. The restoration of PS normalized AHT in 5-6 and HbA1c in 12-13 weeks.
Authors demonstrated the model, which fits obtained experimental data. According to the model, pathways of onset and recovery from pre-DM are different. The consequence of these differences is discussed. The great significance of TCAAEBC for medical practice forces to create appropriate mathematical model, but the required adjustment of the model to needs experimental data, which could be obtained only from animal model. The essential part of this study is devoted to the analysis of advantages and disadvantages of wide available common mammalian models for TCAAEBC case.
Finally, authors conclude that the onset and recovery itineraries from pre-DM are different; and also that the slope of HbA1c dependence from PS demonstrates two different areas.
Overall, the manuscript is valuable for the scientific community and should be accepted for publication after edits are made.
===========================
Other comments:
1) Please check for typos throughout the manuscript.
2) With regards to diabetes mellitus – authors are kindly encouraged to cite the following article that describes the novel approaches of targeting autoimmune diseases such as diabetes mellitus. DOI: 10.1007/s12668-016-0233-x
Please improve the quality of English and the overall text integrity
Author Response
2023-07-25
To Reviewer 2
Dear Reviewer:
Thank you so much for you high evaluation of our humble contribution as well as for your efforts to improve its readability. As about your comments, let me address them in the order in your review:
1) Please check for typos throughout the manuscript.
We carefully edited the entire body of submission
2) With regards to diabetes mellitus – authors are kindly encouraged to cite the following article that describes the novel approaches of targeting autoimmune diseases such as diabetes mellitus. DOI: 10.1007/s12668-016-0233-x
Done.
Please, let us know if we can do something else to improve our submission.
Regards
Dr. Alexandre A. Vetcher
Round 2
Reviewer 1 Report
The authors have responded satisfactorily to my comments, congratulations.